# No Existence and Smoothness of Solution of the Navier-Stokes Equation

**DOI:** 10.3390/e24030339

**Published:** 2022-02-26

**Authors:** Hua-Shu Dou

**Affiliations:** Faculty of Mechanical Engineering and Automation, Zhejiang Sci-Tech University, Hangzhou 310018, China; huashudou@zstu.edu.cn

**Keywords:** Navier-Stokes equation, singularity, transitional flow, turbulence, Poisson equation

## Abstract

The Navier-Stokes equation can be written in a form of Poisson equation. For laminar flow in a channel (plane Poiseuille flow), the Navier-Stokes equation has a non-zero source term (∇^2^*u*(*x, y, z*) = *F_x_* (*x, y, z, t*) and a non-zero solution within the domain. For transitional flow, the velocity profile is distorted, and an inflection point or kink appears on the velocity profile, at a sufficiently high Reynolds number and large disturbance. In the vicinity of the inflection point or kink on the distorted velocity profile, we can always find a point where ∇^2^*u*(*x, y, z*) = 0. At this point, the Poisson equation is singular, due to the zero source term, and has no solution at this point due to singularity. It is concluded that there exists no smooth orphysically reasonable solutions of the Navier-Stokes equation for transitional flow and turbulence in the global domain due to singularity.

## 1. Introduction

In the past 50 years, researchers have conducted theoretical, experimental and direct numerical simulation (DNS) works on the Navier-Stokes equation and have shown that the flow field governed by this equation coincides well with the data on both the laminar flow and the turbulent flow. Therefore, people believe that the Navier-Stokes equation describes both the laminar flow and turbulence qualitatively and quantitatively. However, whether the three-dimensional (3D) incompressible Navier-Stokes equation has unique smooth (continuously differentiable) solutions is still not known [1,2].

Leray showed that the Navier-Stokes equations in three space dimensions always have a weak solution for velocity and pressure, with suitable growth properties [3], but the uniqueness of weak solutions of the Navier-Stokes equation is not demonstrated. Further, the existence of a strong solution (continuously differentiable) of the Navier-Stokes equations is still a challenge in the community of mathematics and physics, although much effort has been made around the world.

Dou and co-authors studied the origin of turbulence using the energy gradient theory [4,5,6,7,8,9] and discovered that there is velocity discontinuity in transitional flow and turbulence [9], which is a singularity of the Navier-Stokes equation. The singularity found theoretically is in agreement with the burst phenomenon in experiments. It was concluded that there exist no smooth and physically reasonable solutions of the Navier-Stokes equation at a high Reynolds number (beyond laminar flow) [9].

As is well known, the flow of viscous incompressible fluid is governed by the Navier-Stokes equation, which is a Poisson equation. The steady laminar flow is dominated by the Poisson equation with the source term of no vanishing. As observed in experiments and simulations, when the incoming laminar flow is disturbed by nonlinear disturbance, the velocity profile is distorted at a sufficient high Reynolds number. In the distorted flow, there may be some points on the velocity profile where the source term becomes zero, which form singularities of the corresponding Poisson equation. The existence of these singular points may lead to no solution of the Navier Stokes equation.

Singularity of the Navier-Stokes equation has received extensive study, owing to its importance in partial differential equations and turbulence [1]. In the literature, there are two different types of singularities described. These singularities are both located off the solid walls. The first type is the one formed by the unbounded kinetic energy of fluid in the flow field [1,3]. The second type is defined at the location where the streamwise velocity of fluid is theoretically zero [9]. The formation mechanisms of these two kinds of singularities are completely different. The former is caused by local infinite acceleration of fluid, and finally blowing up takes place. The latter is resulted from the variation of the velocity profile caused by disturbance in the flow field, which is the singularity of the Navier-Stokes equation itself at some location. This kind of singularity can only occur in viscous flow and does not occur in inviscid flow. In contrast, the first type of singularity may occur in inviscid flows [10]. It has been shown that the first type of singularity may be formed via reconnection of vortex rings in viscous flows [11,12].

In this study, the behavior of the Navier-Stokes equation in the Poisson equation form in transitional flow and turbulence is studied by analyzing the evolution of the velocity profile under finite disturbance, and the singular point of the Poisson equation is explored in the flow domain. No existence and smoothness of solution of the Navier-Stokes equation is concluded for transitional flow and turbulence.

Moffatt has restated the well-known Clay millennium prize problem essentially as this [13]: “can any initially smooth velocity field offinite energy in an incompressible fluid become singularat finite time under Navier-Stokes evolution?” The answer from the reasoning in present study is certainly, if the Reynolds number is sufficiently high and the disturbance is sufficiently large to lead to velocity deficit.

## 2. Stability and Turbulent Transition of Plane Poiseuille Flow

The three-dimensional laminar flow between two parallel walls is as shown in Figure 1 (plane Poiseuille flow). The width of space between two plates in the spanwise direction is infinite. The height in wall-normal direction between the two plates is 2h. The wall is set as the no-slip condition. The incoming flow is a laminar velocity profile. The downstream boundary is set as the Neumann boundary condition. The exact solution of the velocity for the laminar flow is a parabolic velocity distribution along the height for Newtonian fluid [14]. This smooth velocity distribution is placed in the flow field as the initial condition. Then, we observe the variation of the velocity distribution with time under finite disturbances, as in simulations and experiments [15,16,17,18].

With the flow development from the interaction of the base flow with the disturbance, the velocity profile can be modified, depending on the Reynolds number and the disturbance, as in simulations and experiments [15,16,17,18]. For the incoming laminar velocity profile (Figure 2a) at a sufficiently high Reynolds number, after the velocity profile is distorted, inflection point appears first (point A in Figure 2b), and then a section with positive second derivatives appears on the downstream velocity profile (section A–B in Figure 2c). The velocity profile with positive second derivatives may play an important role in the formation of singularity. Three features of the streamwise velocity profile are shown in Figure 2.

Numerical simulations and experimental data show that when the laminar flow is disturbed, the velocity profile will change, and some positions of the velocity profile will be distorted. The results of theoretical analysis on plane Poiseuille flow by Dou show that the basic flow has the maximum ability to amplify the disturbance at *y*/*h* = ±0.58, and the velocity distortion is the largest there [4,5]. Numerical calculations and experiments have shownthat the place where the maximum disturbance appears and the velocity profile change first occurs is at *y*/*h* = ±0.58 [15], where the velocity profile shows an inflection point. These results confirmed the analytical results by Dou and co-authors [4,5,6,7,8,9]. However, there is little change in the velocity profile at the center line and near the two walls in the early stage of disturbance amplification in plane Poiseuille flow.

Dou proved with the energy gradient theory that when there is an inflection point on the velocity profile, discontinuity (negative spike) of streamwise velocity occurs in the temporal evolution under disturbance [9], which is in agreement with simulations and experiments. A model for the velocity distribution at the discontinuity was proposed as shown in Figure 3, which occurs immediately after the inflection point is formed on the velocity profile.

Leray did pioneering work on the weak solution of the Navier-Stokes equations [3]. Foias et al. summarized that [1]: “Leray speculated that turbulence is due to the formation of point or ‘line vortices’ on which some component of the velocity becomes infinite.” “Even today, despite much effort, Leray’s conjecture concerning the appearance of singularities in 3-dimensional turbulent flows has been neither proved nor disproved”.

As far as we know, the singularity in turbulence conjectured by Leray is never found in experiments and simulations (the first type of singularity mentioned in the Introduction). In contrast, the second type of singularity (zero streamwise velocity off the solid wall) is confirmed by experiments and simulation results [19,20,21,22], as described in [9]. The aim of present study is an alternative approach to achieve the same conclusion as that in our previous work [9]: that there exist no smooth and physically reasonable solutions of the Navier-Stokes equation for transitional flow and turbulence in the global domain due to singularity (for pressure driven flows).

## 3. Navier-Stokes Equation in Form of Poisson Equation

### 3.1. Navier-Stokes Equation: Poisson Equation

The continuity and the unsteady momentum equation (Navier-Stokes equation) for incompressible fluid can be written as follows [14]:(1)∇⋅u=0
(2)$$\rho (\frac{\partial u}{\partial t}+u\cdot \nabla u)=-\nabla p+\mu \nabla^{2}u+f$$
where u is the velocity vector, p is the static pressure, ρ is the fluid density, μ is the dynamic viscosity, and f is the gravitational force.

For the pressure-driven flow between two parallel plane walls (Figure 1), the wall boundary condition is no-slip:(3)u=0

Rewriting Equation (2), we have
(4)$$\nabla^{2}u=\frac{1}{\nu }(\frac{\partial u}{\partial t}+u\cdot \nabla u)+\frac{1}{\mu }\nabla p-\frac{1}{\mu }f$$
where ν=μ/ρ is the kinematic viscosity.

Equation (4) is a form of Poisson equation and can be written as follows in Cartesian coordinates:(5)$$\frac{\partial^{2}u}{\partial x^{2}}+\frac{\partial^{2}u}{\partial y^{2}}+\frac{\partial^{2}u}{\partial z^{2}}=F_{x}\left(x,y,z,t\right)$$
$$\frac{\partial^{2}v}{\partial x^{2}}+\frac{\partial^{2}v}{\partial y^{2}}+\frac{\partial^{2}v}{\partial z^{2}}=F_{y}\left(x,y,z,t\right)$$
$$\frac{\partial^{2}w}{\partial x^{2}}+\frac{\partial^{2}w}{\partial y^{2}}+\frac{\partial^{2}w}{\partial z^{2}}=F_{z}\left(x,y,z,t\right)$$
where
(6)$$F_{x}(x,y,z,t)=\frac{1}{\nu }\left(\frac{\partial u}{\partial t}+u\frac{\partial u}{\partial x}+v\frac{\partial u}{\partial y}+w\frac{\partial u}{\partial z}\right)+\frac{1}{\mu }\frac{\partial p}{\partial x}-\frac{1}{\mu }f_{x}$$
$$F_{y}(x,y,z,t)=\frac{1}{\nu }\left(\frac{\partial v}{\partial t}+u\frac{\partial v}{\partial x}+v\frac{\partial v}{\partial y}+w\frac{\partial v}{\partial z}\right)+\frac{1}{\mu }\frac{\partial p}{\partial y}-\frac{1}{\mu }f_{y}$$
$$F_{z}(x,y,z,t)=\frac{1}{\nu }\left(\frac{\partial w}{\partial t}+u\frac{\partial w}{\partial x}+v\frac{\partial w}{\partial y}+w\frac{\partial w}{\partial z}\right)+\frac{1}{\mu }\frac{\partial p}{\partial z}-\frac{1}{\mu }f_{z}$$
where *u*, *v*, and *w* are the velocity components in the *x*, *y*, and *z* direction, respectively.

### 3.2. Reduced Form of Navier-Stokes Equation: Laplace Equation

When the source term in Equation (5) becomes zero in the domain (−h≤y≤+h),
(7)$$F_{x}\left(x,y,z,t\right)\;=0,\;F_{y}\left(x,y,z,t\right)=0,\;F_{z}\left(x,y,z,t\right)=0$$

The Poisson Equation (5) reduces to the Laplace equation form,
(8)$$\frac{\partial^{2}u}{\partial x^{2}}+\frac{\partial^{2}u}{\partial y^{2}}+\frac{\partial^{2}u}{\partial z^{2}}=0,\;\frac{\partial^{2}v}{\partial x^{2}}+\frac{\partial^{2}v}{\partial y^{2}}+\frac{\partial^{2}v}{\partial z^{2}}=0,\;\frac{\partial^{2}w}{\partial x^{2}}+\frac{\partial^{2}w}{\partial y^{2}}+\frac{\partial^{2}w}{\partial z^{2}}=0$$

The solution of the Laplace equation in Equation (8) for plane Poiseuille flow with the no-slip boundary condition at the upper and bottom walls is
(9)u(x,y,z)=0, v(x,y,z)=0, w(x,y,z)=0

This means that the fluid is static.

### 3.3. Solution of Navier-Stokes Equation for Steady Laminar Flow

For the steady parallel laminar flow in plane Poiseuille flow shown as in Figure 1, the pressure gradient in the *x* directionis
(10)∂p/∂x≠0

Then, the source term in Equation (5) is
(11)$$F_{x}\left(x,y,z\right)\neq 0\;(-h\leq y\leq +h)$$

The solution for the Poisson Equation (5) with the no-slip boundary condition is
(12)$$u\left(x,y,z\right)\neq 0\;(-h<y<+h)$$
$$u\left(x,y,z\right)=0\;(y=\pm h)$$

For pressure-driven flow governed by Equation (5), the non-zero solution of velocity is that the source term must not be zero in the Poisson equation (Navier-Stokes equation). However, this conclusion is not true for plane Couette flow (shear driven flow), and a non-zero solution of velocity does not require a non-zero source term in the Poisson equation (Navier-Stokes equation).

The solution of Equation (5) with a source term $F_{x}\left(x,y,z,t\right)=0$ at any location in the domain (−h<y<+h) makes no sense for the studied plane Poiseuille problem, and the position with $F_{x}\left(x,y,z,t\right)=0$ would be a singular point of solution for Equation (12).

As is well known, for steady laminar flow at a low Reynolds number, the solution of Equation (4) or (5) for the parallel flow between two parallel plates (Figure 1) is as follow if the origin of the coordinates is fixed at the centerline [14]
(13)$$u(y)=-\frac{\partial p}{\partial x}\frac{h^{2}}{2\mu }\left(1-\frac{y^{2}}{h^{2}}\right)$$
v=0
w=0

For this solution at low Reynolds number, there is no singular point in such a laminar flow, and the flow is smooth and stable.

## 4. Solution of Navier-Stokes Equation for Transitional and Turbulent Flows

For the laminar flow at a higher Reynolds number in the plane Poiseuille flow configuration, once a disturbance is introduced, the flow becomes time-dependent and three-dimensional. As such, the streamwise velocity is distorted downstream due to the effect of nonlinear disturbance interaction (Figure 2b,c). The velocity components in this three-dimensional flow are v(x,y,z)≠0 and w(x,y,z)≠0, but u>>v and u>>w. In the transitional flow and turbulent flow, at the position of turbulence “burst”, the velocity components *v* and *w* may become large, which are the same order of magnitude as the streamwise component *u*, but the magnitudes of *v* and *w* are still much smaller than the streamwise velocity *u,* as found from previous experiments [19].

### 4.1. Strategy to Solve the Problem

As mentioned before, the existence and smoothness of the Navier-Stokes solution are still unknown. If we find the singularity in the flow field that makes the solution nonexistent in transitional flow and turbulence, we can provide counter evidence to the existence and smoothness of the Navier-Stokes equation.

As discussed for Equation (5), for the steady laminar flow between two parallel plates, $\nabla^{2}u(x,y,z)=F_{x}(x,y,z)$ with $F_{x}(x,y,z)\neq 0$, and the solution within the domain is u(x,y,z)≠0, except at the wall boundaries. There is no singularity in the basic flow here.

In transitional flow, under the nonlinear interaction of disturbance, the velocity profile is distorted at a high Reynolds number, and the distorted velocity profile may produce singularity in temporal evolution. For example, at some point (like the inflection point on velocity profile, which will be shown later) on the velocity profile, the Laplace operator may instantaneously become zero, $\nabla^{2}u(x,y,z)=0$, so that $F_{x}(x,y,z)=0$. At this point, the solution to satisfy the governing equation $\nabla^{2}u(x,y,z)=0$ and the boundary condition u=0 at the wall is u=0.

As shown in Figure 2b,c, the velocity of the given distorted velocity profile is u≠0 except at the wall, while the solution at the inflection point from the governing equation and the boundary condition is u=0 in a time-dependent flow. Thus, this point is a singular point of the Navier-Stokes equation (Equation (5)). Since there exists a singular point in the flow field, there is no smooth solution of the Navier-Stokes equation.

When there is such a singular point, the Navier-Stokes equation has no solution at the singular point. In the transitional flow, we can find such singularity by analyzing the evolution of the velocity profile.

Under the condition at sufficiently high Re and finite disturbance, two types of velocity profiles can be produced, at least as shown in Figure 2b,c. After carrying out analysis, we can always find the point where $\nabla^{2}u(x,y,z)=0$ on these two types of velocity profiles. At such a point, $F_{x}(x,y,z,t)$ = 0, which becomes the singularity of the Poisson equation (Equation (5)).

### 4.2. Finding the Singular Point on the Velocity Profile

The velocity profile of the u component under a disturbance expressed by Equation (5) is shown in Figure 2b,c, while the flow is actually three-dimensional at the transitional Reynolds number. At the inflection point (A in Figure 2b and A and B in Figure 2c), $\frac{\partial^{2}u}{\partial y^{2}}=0$, but $\frac{\partial^{2}u}{\partial x^{2}}+\frac{\partial^{2}u}{\partial z^{2}}$ may not be zero. However, in the vicinity of the inflection point on the velocity profile, we can always find the location where $\frac{\partial^{2}u}{\partial y^{2}}$ and $\frac{\partial^{2}u}{\partial x^{2}}+\frac{\partial^{2}u}{\partial z^{2}}$ have identical magnitude and have opposite signs, since the gradient of flow variables in the y direction is much larger than those in the *x* and *z* directions. Thus, we can have $\nabla^{2}u(x,y,z)=0$ at such a location.

In the following, the singular point that makes the equation $\nabla^{2}u(x,y,z)=0$ established in 3D flows is explored.

(a) When the incoming flow is a laminar velocity profile (two-dimensional), *u* = *u*(*y*), *v* = 0, *w* = 0, we have $\frac{\partial^{2}u}{\partial y^{2}}<0$, $\frac{\partial^{2}u}{\partial x^{2}}=0$, $\frac{\partial^{2}u}{\partial z^{2}}=0$, and $\frac{\partial^{2}u}{\partial x^{2}}+\frac{\partial^{2}u}{\partial y^{2}}+\frac{\partial^{2}u}{\partial z^{2}}<0$ (Figure 2a).

(b) When the velocity is disturbed by finite disturbance, the middle part of the velocity profile is disturbed greatly (becoming three-dimensional), and this part first deforms. Here, it reaches $\frac{\partial^{2}u}{\partial y^{2}}$ = 0 first in the middle part, forming the inflection point A, while the upper and lower parts of the velocity profile do not change much (Figure 2b). In the following, we will discuss these as two different cases, respectively.

(c) If $\frac{\partial^{2}u}{\partial x^{2}}+\frac{\partial^{2}u}{\partial z^{2}}$ > 0 at point A (noting $\frac{\partial^{2}u}{\partial y^{2}}=0$ at point A), then $\frac{\partial^{2}u}{\partial x^{2}}+\frac{\partial^{2}u}{\partial y^{2}}+\frac{\partial^{2}u}{\partial z^{2}}$ > 0.

Observing Figure 2b, in the top and bottom parts of the velocity profile, D1 and D2 locations, the disturbance is small, and there still exist $\frac{\partial^{2}u}{\partial x^{2}}\approx 0$, $\frac{\partial^{2}u}{\partial z^{2}}\approx 0$, $\frac{\partial^{2}u}{\partial y^{2}}$ < 0, $\left|\frac{\partial^{2}u}{\partial y^{2}}\right|$>> $\left|\frac{\partial^{2}u}{\partial x^{2}}+\frac{\partial^{2}u}{\partial z^{2}}\right|$; thus, $\frac{\partial^{2}u}{\partial x^{2}}+\frac{\partial^{2}u}{\partial y^{2}}+\frac{\partial^{2}u}{\partial z^{2}}$ < 0. Then, it is necessary that there exist C1 and C2 points between A and D1 and between A and D2, respectively, which makes $\nabla^{2}u(x,y,z)=0$.

(d) If $\frac{\partial^{2}u}{\partial x^{2}}+\frac{\partial^{2}u}{\partial z^{2}}$ < 0 at point A (noting $\frac{\partial^{2}u}{\partial y^{2}}=0$ at point A), then $\frac{\partial^{2}u}{\partial x^{2}}+\frac{\partial^{2}u}{\partial y^{2}}+\frac{\partial^{2}u}{\partial z^{2}}$ < 0.

Observing Figure 2c, the velocity profile develops, reaching the stage of section A–B where $\frac{\partial^{2}u}{\partial y^{2}}$ > 0 (section A–B), while $\frac{\partial^{2}u}{\partial x^{2}}+\frac{\partial^{2}u}{\partial z^{2}}$ < 0. With the evolution of the velocity profile, the value of $\frac{\partial^{2}u}{\partial y^{2}}$ at C point increases gradually from zero to positive. When the magnitude of $\frac{\partial^{2}u}{\partial y^{2}}$ equals that of $\frac{\partial^{2}u}{\partial x^{2}}+\frac{\partial^{2}u}{\partial z^{2}}$, then we have $\nabla^{2}u(x,y,z)=0$ at one point in the A–B section.

Therefore, when an inflection point is formed on the velocity profile, a section with a positive value of $\frac{\partial^{2}u}{\partial y^{2}}$ will be formed inevitably, with further evolution under a sufficiently large disturbance at a sufficiently high Reynolds number. As long as the positive value of $\frac{\partial^{2}u}{\partial y^{2}}$ is sufficiently large at this section (A–B) on the velocity profile and is able to offset the value of $\frac{\partial^{2}u}{\partial x^{2}}+\frac{\partial^{2}u}{\partial z^{2}}$, there always exists a location within the A–B section with $\nabla^{2}u(x,y,z)=0$.

### 4.3. Solution with Variation of Source Term

The following analysis can be made for various magnitudes of the source term in Equation (5):

(1) $F_{x}(x,y,z,t)$ = 0, Equation (5) becomes the Laplace equation, Equation (7), and the solution is u(x,y,z)=0, v(x,y,z)=0, and w(x,y,z)=0.

(2) $F_{x}(x,y,z,t)$ is small, i.e., Re is low, the nonlinear disturbance effect is small in the base laminar flow, and the velocity profile is not distorted downstream. Finally, the disturbance is damped, and the flow stays laminar.

(3) $F_{x}(x,y,z,t)$ is large, i.e., Re is high, the nonlinear disturbance effect is larger, and the velocity profile is distorted downstream, which leads to a transitional flow. The inflection point or kink appears on the velocity profile in the transitional flow. There is a position in the vicinity of the inflection point (A) where $\nabla^{2}u(x,y,z)=0$ is established (Figure 2b,c). This position becomes the singular point of the Poisson equation (Equation (5)) due to $F_{x}(x,y,z,t)$ = 0 at this point. Figure 4 shows the schematic of the solution strategy of the disturbed flow.

In Figure 4a, the base flow is defined by the Poisson equation (Navier–Stokes), $\nabla^{2}u(x,y,z)=F_{x}(x,y,z,t)$, and the source term is not zero, $F_{x}(x,y,z,t)\neq 0$. The solution of the governing equation with the wall no-slip boundary conditions is u(x,y,z)≠0, except at the walls.

In Figure 4b, after the base flow is disturbed, the velocity profile is distorted locally. The governing equation is still the Poisson equation (Navier–Stokes), $\nabla^{2}u(x,y,z)=F_{x}(x,y,z,t)$, and the source term is still not zero, $F_{x}(x,y,z,t)\neq 0$. However, there is an exception at the inflection point A or its neighborhood, $\nabla^{2}u(x,y,z)=0$ (i.e., $F_{x}(x,y,z,t)=0$), which is singular in the flow field. Thus, there is no solution for Equation (5) at point A.

In Figure 4c, for flow field controlled by $\nabla^{2}u(x,y,z)=0$ and the wall no-slip boundary condition, the solution of flow field is u(x,y,z)=0. Thus, the value of the streamwise velocity at point A in Figure 4b should be zero, as discussed for Equations (7)–(9).

Thus, after inflection points are formed in Figure 4b, the velocity at point A in Figure 4b will theoretically become zero immediately at the next moment in the temporal evolution. This is shown in Figure 5. Because of the viscosity of fluid, the velocity at point A will not be absolutely zero, but spikes are produced, as shown in Figure 3. Simultaneously, fluctuations of the velocity components *v* and w as well as the pressure *p* are produced, which follow the conservation of the total mechanical energy before and after the spike generation. At the inflection point or its neighborhood, where the spike is produced, the fluid element is compressed in the streamwise direction, and thus it is stretched in wall-normal and spanwise directions. Since the mainstream velocity decreases, the pressure will increase at the said singular location. Therefore, the fluctuation of *u* is firstly negative, and those of *v**, w* and *p* are positive, at the singular point. These variations of fluctuations of velocity components and pressure are in agreement with the experimental results of turbulent burst in plane Poiseuille flow [19,23].The feature of positive pressure maximum associated with the burst of turbulence has also been found in the boundary layer flow on flat plates [24,25,26].

Swearingen et al. found through experiments and simulations for wall-bounded flow that the turbulence production events are preceded by an inflectional velocity profile [27]. In bounded transitional flows, this unstable profile produces velocity fluctuations in the streamwise direction and in the other two directions. Figure 5 provides a theoretical interpretation for the generation of fluctuations in wall-bounded transitional flows (pressure driven flow). The result shown in Figure 5 is in agreement with the experimental and simulation results in [19,20,21,22].

In Figure 4b, since point A is singular, the velocity is not differentiable at this position (as shown in Figure 3 and Figure 5). Therefore, there is no smooth solution of the Navier-Stokes equation (Equation (5)) at point A in the transitional flow (Figure 4b).

(4) For fully developed turbulent flow, $F_{x}(x,y,z,t)$ is large, i.e., Re is very high. The velocity profile is heavily distorted by the vortex overlap to the streamwise velocity profile [9], and velocity profiles with an inflectional point or kink are formed, which leads to $\nabla^{2}u(x,y,z)=0$(i.e.,$F_{x}(x,y,z,t)$ = 0) at these points. As such, there is a lot of singular points of Equation (5), as expressed in Figure 2b,c. Therefore, there exist no continuous smooth solutions of Equation (5) for turbulent flow in the global domain.

Fletcher has discussed the characteristics of the general Poisson equation, where the source term can be set up as any value [28]. That is, there is no limitation on the value of the source term, and the zero source term can be defined anywhere in the flow field. For example, for the thermal conduction problem between two parallel plates, the source function of the resulting Poisson equation can be taken as any value (including zero value), and the solution of temperature function has no singularity within the domain.

For plane Poiseuille flow, the governing equation (Navier-Stokes equation) can be written as a form of Poisson equation, as in Equation (5). According to the Navier-Stokes equation and the boundary conditions of plane Poiseuille flow, the source term of the Poisson equation is not arbitrary and it must not be zero. Thus, the zero value of the source term of Equation (5), if any, is not defined in the domain. At any position in the flow field, as long as the source term is zero, it constitutes the singularity of the Navier-Stokes equation, which makes the equation have no solution.

## 5. Significance of the Singularity in Turbulence

In the transition of laminar flow to turbulence, the singularity at the inflection point (or its neighborhood) results in a “burst”, as observed by previous experiments [9]. The burst is the origin of turbulence generation and production of turbulent stresses. In other words, the flow relieves this singularity by a “burst” at the inflection point, and the singularity is converted into turbulent fluctuations at the said position.

In present study, singularity on the velocity profile in finite time for the incompressible Navier-Stokes equations is both mathematical and physical. In mathematics, this singularity of the Navier-Stokes equation occurs at (or near) the inflection point, which makes the equation not differentiable at this position. In physics, this singularity leads to a “burst” as well as fluctuations of streamwise velocity and other velocity components, i.e., turbulence if the Reynolds number is sufficiently high. The mechanical energy of the mainstream flow is transmitted to turbulent fluctuations via this singularity. The singularity is also the reason why turbulence cannot be repeated exactly.

In physical fluid flows in the laboratory, the flow is three-dimensional under finite disturbance, rather than two-dimensional. Further, two-dimensional disturbance isn’t able to induce this type of singularity, since the fluid element is subjected to both shear and extension at the singularity. In other words, there is also stretch of spanwise vorticity at the singular point.

As discussed before, at the singular point (inflection point), the streamwise velocity is theoretically zero. In order to conserve the total mechanical energy, the pressure at this point reaches its maximum. In the near upstream of the singular point, the disturbance is already three-dimensional (even though the amplitude is small). At the singular point, the disturbance is largest, with that singularity leading to “explosive burst” where the amplitudes of fluctuations of both velocity and pressure are largest. The pressure maximum at the singular point produces a pressure wave that spreads as its elliptical property.

Finally, it is pointed out that since the studied singularity of the Navier-Stokes equation is caused by a vanishing viscous term in the Navier-Stokes equation at the inflection point of the velocity profile in the flow domain (Laplace operator is zero), such singularity is never produced in inviscid flows governed by the Euler equation. This implies that turbulence, the properties of which are dissipative, with temporal bursting, with increasing resistance, and with self-sustained fluctuations, could not be generated in inviscid flows.

## 6. Conclusions

Solutions of the Navier-Stokes equation with the Poisson equation form are studied by analyzing the variation of the velocity profile versus the Re number and the disturbance. For the steady laminar flow between two parallel plates, the Poisson equation dominates the flow with the source term of no vanishing. For the laminar flow at a sufficiently high Reynolds number and under certain finite disturbance, the velocity profile is distorted downstream and an inflection point appears (or kink appearance). With the evolution of the velocity profile under finite disturbance, in the vicinity of the inflection point, it is found that there is always a position with $\nabla^{2}u(x,y,z)=0$ (i.e.,$F_{x}(x,y,z,t)$ = 0). This point is singular in the global domain for the Poisson equation (Navier-Stokes equation). At this kind of singular point, the flow variables are not differentiable. Therefore, there exist no smooth and physically reasonable solutions of the Navier-Stokes equation in transitional and turbulent flows.

It should be pointed out that the reasoning presented in this study is only for pressure-driven flows. For shear driven flow, the same conclusion on existence and smoothness of the solution of the Navier-Stokes equation can be obtained with the boundary conditions changed, but the work done by shear stress should be taken into account (this work will be published in separate paper). For shear-driven flows, the mechanisms of instability occurrence and turbulent transition have been studied, respectively, for plane Couette flow and Taylor-Couette flow in [7,8], where external work has been included.

The above conclusion confirmed the analysis results with the energy gradient theory in [9], which show occurrences of streamwise velocity suddenly stop and velocity discontinuity due to zero mechanical energy drop along the streamline. It was shown that the discontinuity of streamwise velocity forms the singularity of the Navier-Stokes equation.

Therefore, both approaches using energy gradient theory and Poisson equation analysis are consistent and show that there is a singular point in the vicinity of the inflection point on the velocity profile where the streamwise velocity is theoretically zero. Neither existence nor smoothness of the solution of the Navier-Stokes equation is demonstrated for transitional and turbulent flows.

## Figures and Tables

**Figure 1 entropy-24-00339-f001:**
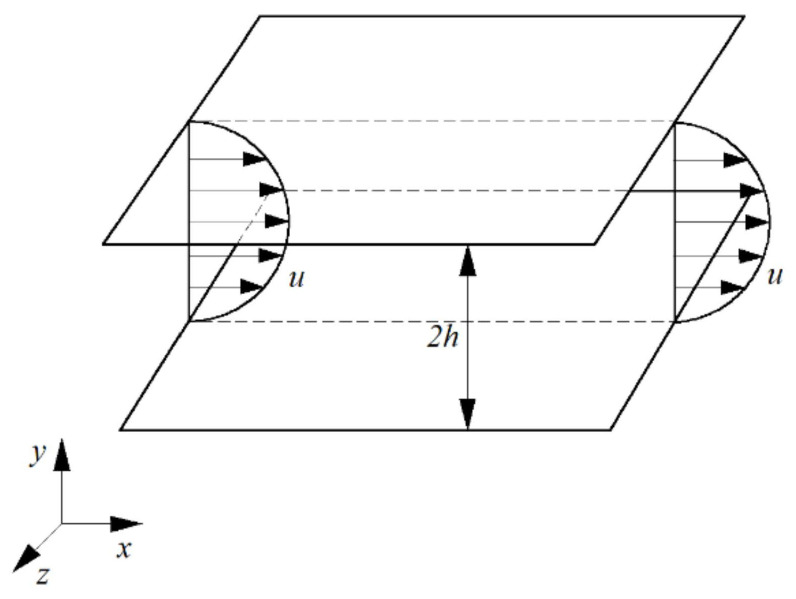
Plane Poiseuille flow between two parallel plates with boundary conditions and initial conditions.

**Figure 2 entropy-24-00339-f002:**
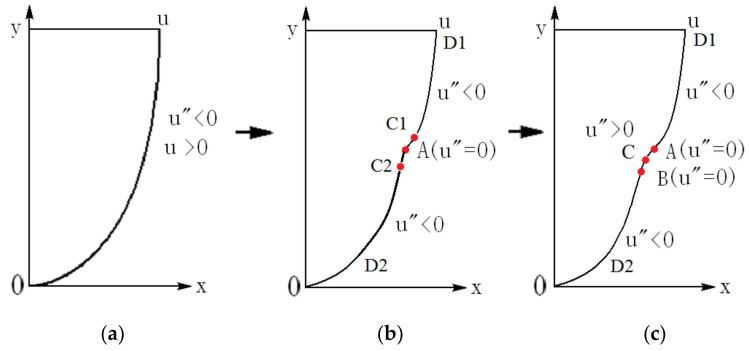
(**a**) Velocity profile of laminar flow; (**b**) an inflection point appears on the velocity profile indicated by A, and the second derivative of velocity $u^{\prime \prime }<0$, except point A; (**c**) the second inflection point B is produced after the first inflection point A, and a section of $u^{\prime \prime }>0$ appears on the velocity profile (A–B section). Here, $u^{\prime \prime }$ stands for the second derivative of the velocity to the direction normal to the wall, $\partial^{2}u/\partial y^{2}$. In the figure, A: inflection point; B: second inflection point; D1: upper part; D2: lower part; C1: point between A and D1 in (**b**); C2: point between A and D2 in (**b**); C: point between A and B in (**c**).

**Figure 3 entropy-24-00339-f003:**
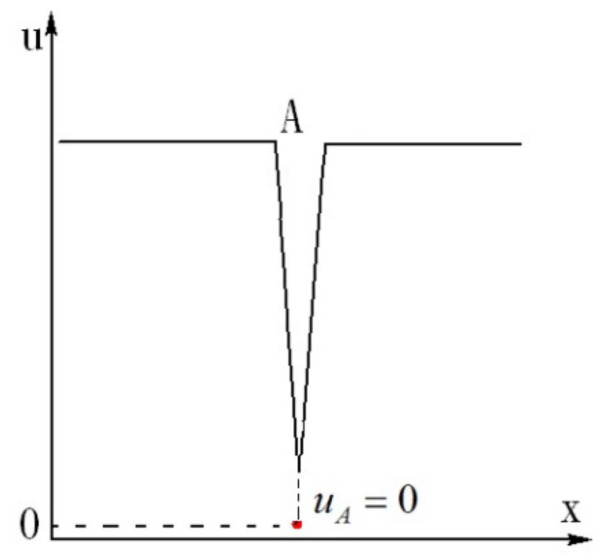
Streamwise velocity distribution under finite disturbance for high Reynolds number flow (transitional flow), showing the singular point (velocity discontinuity) in the vicinity of the inflection point A [9].

**Figure 4 entropy-24-00339-f004:**
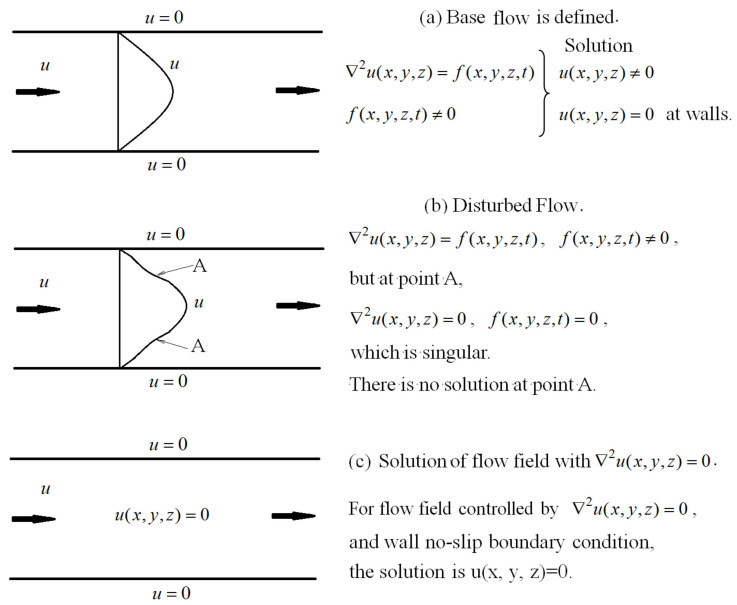
Solution of streamwise velocity showing that there is a singular point for the disturbed velocity distribution.

**Figure 5 entropy-24-00339-f005:**
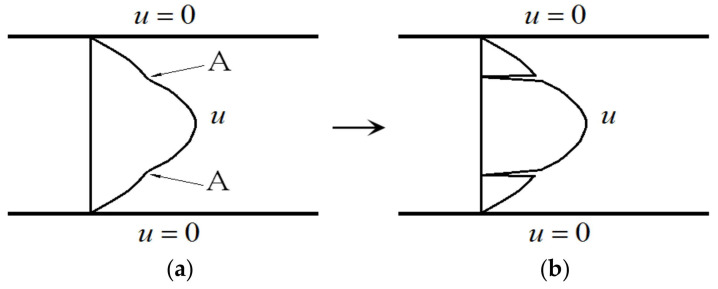
Temporal evolution of velocity profile in transitional flow. (**a**) Velocity profile with inflection points. (**b**) Singular points appear theoretically with *u* = 0 at the locations of zero source term.

## Data Availability

The data are contained within the article.

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
