# Peer review of "No Existence and Smoothness of Solution of the Navier-Stokes Equation"

_entropy, 2022, doi:10.3390/e24030339_

Round 1
Reviewer 1 Report
This review is about the paper entitled as "No Existence and Smoothness of Solution of the Navier-Stokes Equation", Manuscript ID: entropy-1584759.
In this paper, the author concluded that there's no smooth solution of the Navier-Stokes in transitional and turbulent flows due to singularity. The whole study is based on various assumptions of the Laplacian of the velocity components in three dimensions and the second derivative of velocity in vertical direction. The work is confined in plane Poiseuille profiles.
This study is purely theoretical and concerns only pressure driven flows. I am not excited about this paper, but it may be interesting for the readers of the journal. However the author should extend the study deeper in more complex flows which I think is a missing point. As I didn't find anything wrong the way the paper is, I recommend acceptance.
Reviewer 2 Report
Review of the article No Existence and Smoothness of Solution of the Navier-Stokes Equation
The paper is theoretical in nature and deals mainly with the dilemma of Poisson and the Laplace equations and inflection point. I have some remarks about this paper.
- The form of presentation, description. Personally, I prefer to use the passive side. Of course, it is an individual matter and resulting from the style of the author.
- What is the purpose of the article? The issues raised are obvious and not new in the scientific world. Is the purpose of the article to be discussative, descriptive, or analytical?
- There is no evidence presented in the paper, no parametric studies. Maybe it is worth including such analyses in the article, showing that this problem is relevant to such and not other dimensionless numbers.
- The paper will be published in a journal called Entropy. What is the link between the content of the article and the word entropy?
Reviewer 3 Report
The subject matter is a solution to the incompressible Navier-Stokes grand challenge for blowing up in finite time. The author has argued cogently for the occurrence of a singularity in transition from parabolic laminar flow to turbulence due to the inflection point. The mathematical formulation is simple and clear, with the argument followed through logically to show that inflection leads to a singularity of the incompressible Navier-Stokes equations when formulated as a Poisson equation -- the solution at the inflection point is instantaneously zero axial velocity, so discontinuous and non-differentiable.
The paper leaves the reader, however, with the mystery, "What actually happens, physically, to relieve this singularity." The author cites experimental evidence that the orthogonal velocity components become quite large, but obviously no black hole is formed in transition (or every passing wind would generate a black hole!)
There are at least two competing theories. Several years ago, I had a discussion with the late Professor CI Christov, who posited that the question of singularity in finite time for the incompressible Navier-Stokes equations is of mathematical interest only, as they are unphysical. I agreed, but we had different reasons for their being unphysical:
Christov: The viscous constituency equation for a Newtonian fluid is subtly incorrect. There is a weak hyperviscosity term that would enter the momentum equations at 4th order.
See, for instance, for a definition of a hyperviscosity
Abdelhafid Younsi. Effect of hyperviscosity on the Navier-Stokes turbulence. Electronic Journal of
Differential Equations, Texas State University, Department of Mathematics, 2010, 2010 (110), pp.19ff. hal-00459978v2f
https://hal.archives-ouvertes.fr/hal-00459978/document
Me: There is no thermodynamics in the incompressible NS equations -- but no fluid is incompressible. Even common liquids are weakly compressible, so that the incompressible NS equations discards sounds waves. I have (unpublished) experiments in liquid where the discrepancy between the laminar, incompressible NS predictions and the actual pressure measurements, time resolved and high precision, are large, and you can hear the sound waves radiating mechanical energy. It occurs to me that our flowfield has inflection points by construction, but the discrepancy occurs upstream of the inflection points.
I think you are in position to discuss both of these potentially regularizing mechanisms for the singularity induced by inflection points that you have identified.
1. Hyperviscosity, at least to me, is a difficult concept because, as an engineer, I tried to imagine the circumstances where it could be measured, as it must be a small effect relative to Newtonian viscosity. Since all viscometric flows result in the sufficient characterization of Newtonian viscosity, and they are all regular and differentiable, finding a constructive measurement of hyperviscosity would require that the Laplacian of the velocity field vanish. Hence it can only be inferred from an inflection point. I only realized this while reading your manuscript. I have been stumped for over a decade as to how to infer hyperviscosity. You have answered the key feature of a hyper-viscometric flow!
2. I think you are also in a position to address whether sound waves would radiate from the infection point in your transitional flow. Relieving constraint (1) and replacing it with the material derivative of density is zero, does not change any of your arguments about the existence of an inflection point creating a singularity. With the additional degree of freedom of sound waves, it seems quite likely that a singularity would excite them, so there should be an acoustic response / signature for an inflection point, but they do not relieve the singularity you identified in the NS equations, at least if I follow your arguments correctly. rho = rho (t) changes qualitatively none of the arguments for the Poisson equation becoming singular due to an instantaneous inflection point.
Should you wish to correspond with me about either point, I will indicate to the editor that I am willing to waive anonymity.
The English composition could use some copy-editing, but the only point where it makes a material difference to the communication of the concepts is line 203 where "opposite symbol" should be "opposite sign" or "opposite sense" to convey the correct meaning.
In line 304, "the flow variables are not differential", I believe, has the intent "the flow variables are not differentiable", but the meaning is clear.
Otherwise, I am satisfied that this paper is a substantial contribution to the understanding of theoretical fluid mechanics, and should be published with only minor English language changes. I would love to see an expanded discussion to address the points I made above, but it is the author's paper to publish!
Reviewer 4 Report
Please, see the attached review.

Round 2
Reviewer 2 Report
In my opinion, the article can be proceeded with for further publication procedures.
Yours sincerely
Reviewer
Author Response
Thank you for your comments.
Reviewer 4 Report
Please, see the attached report.
